# Learning to Orient Surfaces
# by Self-supervised Spherical CNNs

**Riccardo Spezialetti**[1]**, Federico Stella**[1]**, Marlon Marcon**[2]**, Luciano Silva**[3]

**Samuele Salti** [1]**, Luigi Di Stefano** [1]
[1] Department of Computer Science and Engineering (DISI), University of Bologna, Italy
[2] Federal University of Technology - Paraná, Dois Vizinhos, Brazil
[3] Federal University of Paraná, Curitiba, Brazil
[1]`{riccardo.spezialetti, federico.stella5}@unibo.it`
[2]`marlonmarcon@utfpr.edu.br`
[3]`luciano@inf.ufpr.br`

## Abstract

Defining and reliably finding a canonical orientation for 3D surfaces is key to many Computer Vision and Robotics applications. This task is commonly addressed by handcrafted algorithms exploiting geometric cues deemed as distinctive and robust by the designer. Yet, one might conjecture that humans learn the notion of the inherent orientation of 3D objects from experience and that machines may do so alike. In this work, we show the feasibility of learning a robust canonical orientation for surfaces represented as point clouds. Based on the observation that the quintessential property of a canonical orientation is equivariance to 3D rotations, we propose to employ Spherical CNNs, a recently introduced machinery that can learn equivariant representations defined on the Special Orthogonal group SO(3). Specifically, spherical correlations compute feature maps whose elements define 3D rotations. Our method learns such feature maps from raw data by a self-supervised training procedure and robustly selects a rotation to transform the input point cloud into a learned canonical orientation. Thereby, we realize the first end-to-end learning approach to define and extract the canonical orientation of 3D shapes, which we aptly dub *Compass*. Experiments on several public datasets prove its effectiveness at orienting local surface patches as well as whole objects.

## 1  Introduction

Humans naturally develop the ability to mentally portray and reason about objects in what we perceive as their *neutral*, *canonical* orientation, and this ability is key for correctly recognizing and manipulating objects as well as reasoning about the environment. Indeed, mental rotation abilities have been extensively studied and linked with motor and spatial visualization abilities since the 70s in the experimental psychology literature [36, 40, 16].

Robotic and computer vision systems similarly require neutralizing variations w.r.t. rotations when processing 3D data and images in many important applications such as grasping, navigation, surface matching, augmented reality, shape classification and detection, among the others. In these domains, two main approaches have been pursued so to define rotation-invariant methods to process 3D data: rotation-invariant operators and canonical orientation estimation. Pioneering works applying deep learning to point clouds, such as PointNet [31, 32] achieved invariance to rotation by means of a transformation network used to predict a canonical orientation to apply direclty to the coordinates

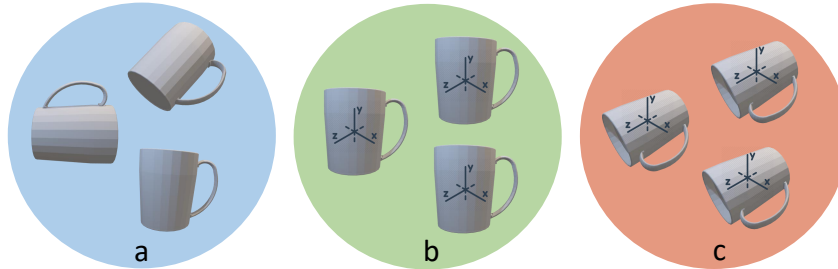

Figure 1: Canonical orientations in humans and machines. Randomly rotated mugs are depicted in (a). To achieve rotation-invariant processing, *e.g.* to check if they are the same mug, humans mentally neutralize rotation variations preferring an *upright* canonical orientation, as illustrated in (b). A machine may instead use any canonical reference orientation, even unnatural to humans, *e.g.* like in (c).

of the input point cloud. Despite being trained by sampling the range of all possible rotations at training time through data augmentation, this approach, however, does not generalize to rotations not seen during training. Hence, invariant operators like rotation-invariant convolutions were introduced, allowing to train on a reduced set of rotations (ideally one, the unmodified data) and test on the full spectrum of rotations [23, 10, 43, 46, 33, 45]. Canonical orientation estimation, instead, follows more closely the human path to invariance and exploits the geometry of the surface to estimate an intrinsic 3D reference frame which rotates with the surface. Transforming the input data by the inverse of the 3D orientation of such reference frame brings the surface in an orientation-neutral, canonical coordinate system wherein rotation invariant processing and reasoning can happen. While humans have a preference for a canonical orientation matching one of the usual orientations in which they encounter an object in everyday life, in machines this paradigm does not need to favour any actual reference orientation over others: as illustrated in Figure 1, an arbitrary one is fine as long as it can be repeatably estimated from the input data.

Despite mental rotation tasks being solved by a set of unconscious abilities that humans learn through experience, and despite the huge successes achieved by deep neural networks in addressing analogous unconscious tasks in vision and robotics, the problem of estimating a canonical orientation is still solved solely by *handcrafted* proposals [35, 28, 24, 12, 42, 11, 1]. This may be due to convnets, the standard architectures for vision applications, reliance on the convolution operator in Euclidean domains, which possesses only the property of *equivariance* to translations of the input signal. However, the essential property of a canonical orientation estimation algorithm is equivariance with respect to 3D rotations because, upon a 3D rotation, the 3D reference frame which establishes the canonical orientation of an object should undergo the same rotation as the object. We also point out that, although, in principle, estimation of a canonical reference frame is suitable to pursue orientation neutralization for whole shapes, in past literature it has been intensively studied mainly to achieve rotation-invariant description of local surface patches.

In this work, we explore the feasibility of using deep neural networks to learn to pursue rotation-invariance by estimating the canonical orientation of a 3D surface, be it either a whole shape or a local patch. Purposely, we propose to leverage Spherical CNNs [7, 10], a recently introduced variant of convnets which possesses the property of equivariance w.r.t. 3D rotations by design, in order to build Compass, a self-supervised methodology that learns to orient 3D shapes. As the proposed method computes feature maps living in $SO(3)$, *i.e.* feature map coordinates define 3D rotations, and does so by rotation-equivariant operators, any salient element in a feature map, *e.g.* its $\arg\max$, may readily be used to bring the input point cloud into a canonical reference frame. However, due to discretization artifacts, Spherical CNNs turn out to be not perfectly rotation-equivariant [7]. Moreover, the input data may be noisy and, in case of 2.5D views sensed from 3D scenes, affected by self-occlusions and missing parts. To overcome these issues, we propose a robust end-to-end training pipeline which mimics sensor nuisances by data augmentation and allows the calculation of gradients with respect to feature maps coordinates. The effectiveness and general applicability of Compass is established by achieving state-of-the art results in two challenging applications: robust local reference frame estimation for local surface patches and rotation-invariant global shape classification.

## 2 Related Work

The definition of a canonical orientation of a point cloud has been studied mainly in the field of local features descriptors [35, 12, 17, 34] used to establish correspondences between sets of distinctive points, *i.e. keypoints*. Indeed, the definition of a robust *local reference frame*, *i.e.* $\mathcal{R}(p) = \{\hat{\mathbf{x}}(p), \hat{\mathbf{y}}(p), \hat{\mathbf{z}}(p) \mid \hat{\mathbf{y}} = \hat{\mathbf{z}} \times \hat{\mathbf{x}}\}$, with respect to which the local neighborhood of a keypoint $p$ is encoded, is crucial to create rotation-invariant features. Several works define the axes of the local canonical system as eigenvectors of the 3D covariance matrix between points within a spherical region of radius $r$ centered at $p$. As the signs of the eigenvectors are not repeatable, some works focus on the disambiguation of the axes [25, 35, 12]. Alternatively, another family of methods leverages the normal to the surface at $p$, *i.e.* $\hat{n}(p)$, to fix the $\hat{\mathbf{z}}$ axis, and then exploits geometric attributes of the shape to identify a reference direction on the tangent plane to define the $\hat{\mathbf{x}}$ axis [6, 29, 24]. Compass differs sharply from previous methods because it learns the cues necessary to canonically orient a surface without making a priori assumptions on which details of the underlying geometry may be effective to define a repeatable canonical orientation.

On the other hand, PointNets[31, 32] employ a transformation network to predict an affine rigid motion to apply to the input point clouds in order to correctly classify global shapes under rigid transformations. In [10] Esteves *et al.* prove the limited generalization of PointNet to unseen rotations and define the Spherical convolutions to learn an invariant embedding for mesh classification. In parallel, Cohen *et al.* [7] use Spherical correlation to map Spherical inputs to $\mathrm{SO}(3)$ features then processed with a series of convolutions on $\mathrm{SO}(3)$. Similarly, PRIN [43] proposes a network based on Spherical correlations to operate on spherically voxelized point clouds. SFCNN [33] re-defines the convolution operator on a discretized sphere approximated by a regular icosahedral lattice. Differently, in [46], Zhang *et al.* adopt low-level rotation invariant geometric features (angles and distances) to design a convolution operator for point cloud processing. Deviating from this line of work on invariant convolutions and operators, we show how rotation-invariant processing can be effectively realized by preliminary transforming the shape to a canonical orientation learned by Compass.

Finally, it is noteworthy that several recent works [26, 41, 38] rely on the notion of canonical orientation to perform category-specific 3D reconstruction from a single or multiple views.

## 3 Proposed Method

In this section, we provide a brief overview on Spherical CNNs to make the paper self-contained, followed by a detailed description of our method. For more details, we point readers to [7, 10].

### 3.1 Background

The base intuition for Spherical CNNs can be lifted from the classical planar correlation used in CNNs, where the value in the output map at location $x \in \mathbb{Z}^2$ is given by the inner product between the input feature map and the learned filter translated by $x$. We can likewise define the value in an output map, computed by a spherical or $SO(3)$ correlation, at location $R \in \mathrm{SO}(3)$ as the inner product between the input feature map and the learned filter *rotated* by $R$. Below we provide formal definitions of the main operations carried out in a Spherical CNN, then we summarize the standard flow to process point clouds with them.

**The Unit Sphere**: $S^2$ is a two-dimensional manifold defined as the set of points $x \in \mathbb{R}^3$ with unitary norm, and parametrized by spherical coordinates $\alpha \in [0, 2\pi]$ and $\beta \in [0, \pi]$.

**Spherical Signal**: a $K$-valued function defined on $S^2$, $f : S^2 \to \mathbb{R}^K$, $K$ is the number of channels.

**3D Rotations**: 3D rotations live in a three-dimensional manifold, the $\mathrm{SO}(3)$ group, which can be parameterized by ZYZ-Euler angles as in [7]. Given a triplet of Euler angles $\alpha, \beta, \gamma$, the corresponding 3D rotation matrix is given by the product of two rotations about the $z$-axis, $R_z(\cdot)$, and one about the $y$-axis, $R_y(\cdot)$, *i.e.* $R(\alpha, \beta, \gamma) = R_z(\alpha)R_y(\beta)R_z(\gamma)$. Points represented as 3D unit vectors $x$ can be rotated by using the matrix-vector product $Rx$.

**Spherical correlation**: recalling the inner product $\langle , \rangle$ definition from [7], the correlation between a $K$-valued spherical signal $f$ and a filter $\psi$, $f, \psi : S^2 \to \mathbb{R}^K$ can be formulated as:

$$[\psi \star f](R) = \langle L_R \psi, f \rangle = \int_{S^2} \sum_{k=1}^{K} \psi_k(R^{-1}x) f_k(x) dx. \tag{1}$$

where the operator $L_R$ rotates the function $f$ by $R \in \mathrm{SO}(3)$, by composing its input with $R^{-1}$, *i.e.* $[L_R f](x) = f(R^{-1}x)$, where $x \in S^2$. Although both the input and the filter live in $S^2$, the spherical correlation produces an output signal defined on $\mathrm{SO}(3)$ [7].

**SO(3) correlation**: similarly, by extending $L_R$ to operate on $\mathrm{SO}(3)$ signals, *i.e.* $[L_R h](Q) = h(R^{-1}Q)$ where $R, Q \in \mathrm{SO}(3)$ and $R^{-1}Q$ denotes the composition of rotations, we can define the correlation between a signal $h$ and a filter $\psi$ on the rotation group, $h, \psi : \mathrm{SO}(3) \to \mathbb{R}^K$:

$$[\psi * h](R) = \langle L_R \psi, h \rangle = \int_{\mathrm{SO}(3)} \sum_{k=1}^{K} \psi_k(R^{-1}Q) h_k(Q) dQ. \tag{2}$$

**Spherical and SO(3) correlation equivariance w.r.t. rotations**: It can be shown that both correlations in (1) and (2) are *equivariant* with respect to rotations of the input signal. The feature map obtained by correlation of a filter $\psi$ with an input signal $h$ rotated by $Q \in \mathrm{SO}(3)$, can be equivalently computed by rotating with the same rotation $Q$ the feature map obtained by correlation of $\psi$ with the original input signal $h$, *i.e.*:

$$[\psi \star [L_Q h]](R) = [L_Q[\psi \star h]](R). \tag{3}$$

**Signal Flow**: In Spherical CNNs, the input signal, *e.g.* an image or, as it is the case of our settings, a point cloud, is first transformed into a $k$-valued spherical signal. Then, the first network layer ($S^2$ layer) computes feature maps by spherical correlations ($\star$). As the computed feature maps are $\mathrm{SO}(3)$ signals, the successive layers ($\mathrm{SO}(3)$ layers) compute deeper feature maps by $\mathrm{SO}(3)$ correlations ($*$).

## 3.2 Methodology

Our problem can be formalized as follows. Given the set of 3D point clouds, $\mathcal{P}$, and two point clouds $\mathcal{V}, \mathcal{T} \in \mathcal{P}$, with $\mathcal{V} = \{p_{\mathcal{V}_i} \in \mathbb{R}^3 \mid p_{\mathcal{V}_i} = (x, y, z)^T\}$ and $\mathcal{T} = \{p_{\mathcal{T}_i} \in \mathbb{R}^3 \mid p_{\mathcal{T}_i} = (x, y, z)^T\}$, we indicate by $\mathcal{T} = R\mathcal{V}$ the application of the 3D rotation matrix $R \in \mathrm{SO}(3)$ to all the points of $\mathcal{V}$. We then aim at learning a function, $g : \mathcal{P} \to \mathrm{SO}(3)$, such that:

$$\mathcal{V}_c = g(\mathcal{V})^{-1} \cdot \mathcal{V} \tag{4}$$
$$g(\mathcal{T}) = R \cdot g(\mathcal{V}). \tag{5}$$

We define the rotated cloud, $\mathcal{V}_c$, in (4) to be the canonical, rotation-neutral version of $\mathcal{V}$, *i.e.* the function $g$ outputs the inverse of the 3D rotation matrix that brings the points in $\mathcal{V}$ into their canonical reference frame. (5) states the equivariance property of $g$: if the input cloud is rotated, the output of the function should undergo the same rotation. As a result, two rotated versions of the same cloud are brought into the same canonical reference frame by (4).

Due to the equivariance property of Spherical CNNs layers, upon a rotation of the input signal each feature map does rotate accordingly. Moreover, the domain of the feature maps in Spherical CNNs is $\mathrm{SO}(3)$, *i.e.* each value of the feature map is naturally associated with a rotation. This means that one could just track any distinctive feature map value to establish a canonical orientation satisfying (4) and (5). Indeed, defining as $\Phi$ the composition of $S^2$ and $\mathrm{SO}(3)$ correlation layers in our network, if the last layer produces the feature map $[\Phi(f_{\mathcal{V}})]$ when processing the spherical signal $f_{\mathcal{V}}$ for the cloud $\mathcal{V}$, the same network will compute the feature map $[L_R \Phi(f_{\mathcal{V}})] = [\Phi(L_R f_{\mathcal{V}})] = [\Phi(f_{\mathcal{T}})]$ when processing the rotated cloud $\mathcal{T} = R\mathcal{V}$, with spherical signal $f_{\mathcal{T}} = L_R f_{\mathcal{V}}$. Hence, if for instance we select the maximum value of the feature map as the distinctive value to track, and the location of the maximum is at $Q_{\mathcal{V}}^{max} \in \mathrm{SO}(3)$ in $\Phi(f_{\mathcal{V}})$, the maximum will be found at $Q_{\mathcal{T}}^{max} = RQ_{\mathcal{V}}^{max}$ in the rotated feature map. Then, by letting $g(\mathcal{V}) = Q_{\mathcal{V}}^{max}$, we get $g(\mathcal{T}) = RQ_{\mathcal{V}}^{max}$, which satisfies (4) and (5). Therefore, we realize function $g$ by a Spherical CNN and we utilize the $\arg\max$ operator on the feature map computed by the last correlation layer to define its output. In principle, equivariance alone would guarantee to satisfy (4) and (5). Unfortunately, while for continuous functions the network is exactly equivariant, this does not hold for its discretized version, mainly due to feature map rotation,

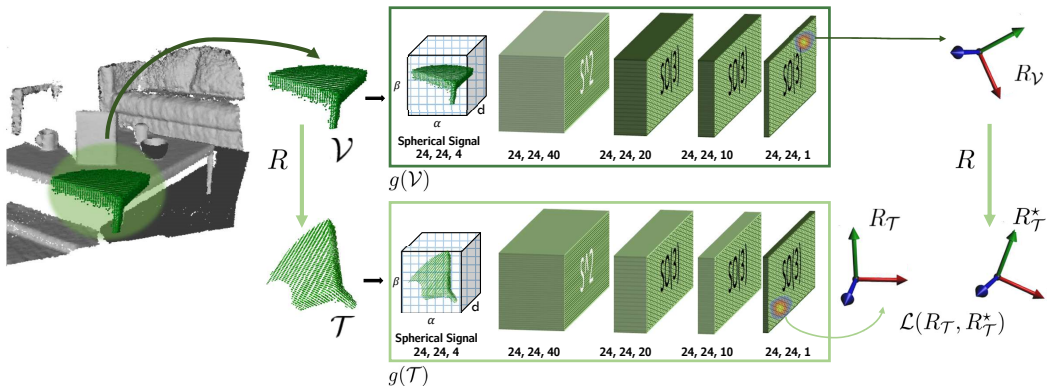

Figure 2: Training pipeline. We illustrate the pipeline for local patches, but the same apply for point clouds representing full shapes. During training we apply the network on a randomly extracted 3D patch, $\mathcal{V}$, and on its augmented version, $\mathcal{T}$, in order to extract the aligning rotation $R_\mathcal{V}$ and $R_\mathcal{T}$, respectively. At test time only one branch is involved. The numbers below the spherical signal indicate the bandwidths along $\alpha$, $\beta$ and $d$, while the triplets under the layers indicate input bandwidth, output bandwidth and number of channels.

which is exact only for bandlimited functions [7]. Moreover, equivariance to rotations does not hold for altered versions of the same cloud, *e.g.* when a part of it is occluded due to view-point changes. We tackle these issues using a self-supervised loss computed on the extracted rotations when aligning a pair of point clouds to guide the learning, and an ad-hoc augmentation to increase the robustness to occlusions. Through the use of a soft-argmax layer, we can back-propagate the loss gradient from the estimated rotations to the positions of the maxima we extract from the feature maps and to the filters, which overall lets the network learn a robust $g$ function.

**From point clouds to spherical signals**: Spherical CNNs require spherical signals as input. A known approach to compute them for point cloud data consists in transforming point coordinates from the input Euclidean reference system into a spherical one and then constructing a quantization grid within this new coordinate system [43, 37]. The $i$-th cell of the grid is indexed by three spherical coordinates $(\alpha[i], \beta[i], d[i]) \in S^2 \times \mathbb{R}$ where $\alpha[i]$ and $\beta[i]$ represent the azimuth and inclination angles of its center and $d[i]$ is the radial distance from the center. Then, the $K$ cells along the radial dimension with constant azimuth $\alpha$ and inclination $\beta$ are seen as channels of a $K$-valued signal at location $(\alpha, \beta)$ onto the unit sphere $S^2$. The resulting $K$-valued spherical signal $f : S^2 \to \mathbb{R}^K$ measures the density of the points within each cell $(\alpha[i], \beta[i])$ at distance $d[i]$.

**Training pipeline**: An illustration of the Compass training pipeline is shown in Figure 2. During training, our objective is to strengthen the equivariance property of the Spherical CNN, such that the locations selected on the feature maps by the $\arg\max$ function vary consistently between rotated versions of the same point cloud. To this end, we train our network with two streams in a Siamese fashion [4]. In particular, given $\mathcal{V}, \mathcal{T} \in \mathcal{P}$, with $\mathcal{T} = R\mathcal{V}$ and $R$ a known random rotation matrix, the first branch of the network computes the aligning rotation matrix for $\mathcal{V}$, $R_\mathcal{V} = g(\mathcal{V})^{-1}$, while the second branch the aligning rotation matrix for $\mathcal{T}$, $R_\mathcal{T} = g(\mathcal{T})^{-1}$. Should the feature maps on which the two maxima are extracted be perfectly equivariant, it would follow that $R_\mathcal{T} = RR_\mathcal{V} = R_\mathcal{T}^\star$. For that reason, the degree of misalignment of the maxima locations can be assessed by comparing the actual rotation matrix predicted by the second branch, $R_\mathcal{T}$, to the ideal rotation matrix that should be predicted, $R_\mathcal{T}^\star$. We can thus cast our learning objective as the minimization of a loss measuring the distance between these two rotations. A natural geodesic metric on the $\mathrm{SO}(3)$ manifold is given by the angular distance between two rotations [13]. Indeed, any element in $\mathrm{SO}(3)$ can be parametrized as a rotation angle around an axis. The angular distance between two rotations parametrized as rotation matrices $R$ and $S$ is defined as the angle that parametrizes the rotation $SR^T$ and corresponds to the length along the shortest path from $R$ to $S$ on the $\mathrm{SO}(3)$ manifold [13, 15, 22, 48]. Thus, our

loss is given by the angular distance between $R_{\mathcal{T}}$ and $R_{\mathcal{T}}^{\star}$:

$$\mathcal{L}(R_{\mathcal{T}}, R_{\mathcal{T}}^{\star}) := \cos^{-1}\left(\frac{(tr(R_{\mathcal{T}}^T R_{\mathcal{T}}^{\star}) - 1)}{2}\right). \tag{6}$$

As our network has to predict a single canonicalizing rotation, we apply the loss once, *i.e.* only to the output of the last layer of the network.

**Soft-argmax**: The result of the $\arg\max$ operation on a discrete $SO(3)$ feature map returns the location $i, j, k$ along the $\alpha, \beta, \gamma$ dimensions corresponding to the ZYZ Euler angles, where the maximum correlation value occurs. To optimize the loss in (6), the gradients w.r.t. the $i, j, k$ locations of the feature map where the maxima are detected have to be computed. To render the $\arg\max x$ operation differentiable we add a soft-argmax operator [14, 3] following the last $SO(3)$ layer of the network. Let us denote as $\Phi(f_{\mathcal{V}})$ the last $SO(3)$ feature map computed by the network for a given input point cloud $\mathcal{V}$. A straightforward implementation of a soft-argmax layer to get the coordinates $C_R = (i, j, k)$ of the maximum in $\Phi(f_{\mathcal{V}})$ is given by

$$C_R(\mathcal{V}) = \text{soft-argmax}(\tau\Phi(f_{\mathcal{V}})) = \sum_{i,j,k}\text{softmax}(\tau\Phi(f_{\mathcal{V}}))_{i,j,k}(i, j, k). \tag{7}$$

where softmax$(\cdot)$ is a 3D spatial softmax. The parameter $\tau$ controls the temperature of the resulting probability map and $(i, j, k)$ iterate over the $SO(3)$ coordinates. A soft-argmax operator computes the location $C_R = (i, j, k)$ as a weighted sum of all the coordinates $(i, j, k)$ where the weights are given by a softmax of a $SO(3)$ map $\Phi$. Experimentally, this proved not effective. As a more robust solution, we scale the output of the softmax according to the distance of each $(i, j, k)$ bin from the feature map $\arg\max$. To let the bins near the $\arg\max$ contribute more in the final result, we smooth the distances by a Parzen function [27] yielding a maximum value in the bin corresponding to the $\arg\max$ and decreasing monotonically to $0$.

**Learning to handle occlusions**: In real-world settings, rotation of an object or scene (*i.e.* a viewpoint change) naturally produces occlusions to the viewer. Recalling that the second branch of the network operates on $\mathcal{T}$, a randomly rotated version of $\mathcal{V}$, it is possible to improve robustness of the network to real-world occlusions and missing parts by augmenting $\mathcal{T}$. A simple way to handle this problem is to randomly select a point from $\mathcal{T}$ and delete some of its surrounding points. In our implementation, this augmentation happens with an assigned probability. $\mathcal{T}$ is divided in concentric spherical shells, with the probability for the random point to be selected in a shell increasing with its distance from the center of $\mathcal{T}$. Additionally, the number of removed points around the selected point is a bounded random percentage of the total points in the cloud. An example can be seen in Figure 3.

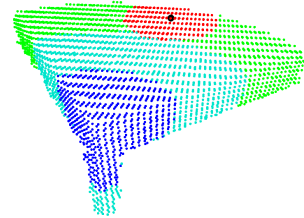

Figure 3: Local support of a keypoint depicting the corner of a table, divided in 3 shells. Randomly selected point in black; removed points in red.

**Network Architecture**: The network architecture comprises 1 $S^2$ layer followed by 3 $SO(3)$ layers, with bandwidth $B = 24$ and the respective number of output channels are set to 40, 20, 10, 1. The input spherical signal is computed with $K = 4$ channels.

## 4 Applications of Compass

We evaluate Compass on two challenging tasks. The first one is the estimation of a canonical orientation of local surface patches, a key step in creating rotation-invariant local 3D descriptors [35, 11, 42]. In the second task, the canonical orientation provided by Compass is instead used to perform highly effective rotation-invariant shape classification by leveraging a simple PointNet classifier. The source code for training and testing Compass is available at `https://github.com/CVLAB-Unibo/compass`.

### 4.1 Canonical orientation of local surface patches

**Problem formulation**: On local surface patches, we evaluate Compass through the repeatability [28, 24] of the local reference frame (LRF) it estimates at corresponding keypoints in different views

Table 1: LRF repeatability on the datasets. Best result for each dataset (row) in bold.

| | | | | | | | Compass |
|---|---|---|---|---|---|---|---|
| | | | LRF Repeatability (*Rep* ↑) | | | | |
| Dataset | SHOT [39] | FLARE [29] | TOLDI [42] | 3DSN [11] | GFrames [24] | Compass | Compass (Adapted) |
| 3DMatch | 0.212 | 0.360 | 0.215 | 0.220 | n.a | **0.375** | n.a. |
| ETH | 0.273 | 0.264 | 0.185 | 0.202 | n.a | 0.308 | **0.317** |
| Stanford Views | 0.132 | 0.241 | 0.197 | 0.173 | 0.256 | 0.361 | **0.388** |

of the same scene. All the datasets provide several 2.5D scans, *i.e.* fragments, representing the same *model*, *i.e.* an object or a scene depending on the dataset, acquired from different viewpoints. All $N$ fragments belonging to a test model can be grouped into pairs, where each pair $(\mathcal{F}_s, \mathcal{F}_t)$, $\mathcal{F}_s = \{p_{s_i} \in \mathbb{R}^3\}$ and $\mathcal{F}_t = \{p_{t_i} \in \mathbb{R}^3\}$, has an area of overlap. A set of correspondences, $\mathcal{C}_{s,t}$, can be computed for each pair $(\mathcal{F}_s, \mathcal{F}_t)$ by applying the known rigid ground-truth transformation, $G_{t,s} = [R_{t,s}|t_{t,s}] \in SE(3)$, which aligns $\mathcal{F}_t$ to $\mathcal{F}_s$ into a common reference frame. $\mathcal{C}_{s,t}$ is obtained by uniformly sampling points in the overlapping area between $\mathcal{F}_s$ and $\mathcal{F}_t$. Finally, the percentage of repeatable LRFs, $Rep_{s,t}$, for $(\mathcal{F}_s, \mathcal{F}_t)$, can be calculated as follows:

$$Rep_{s,t} = \frac{1}{|\mathcal{C}_{s,t}|} \sum_{k=1}^{|\mathcal{C}_{s,t}|} \mathrm{I}\bigg( \big(\hat{\mathbf{x}}(p_{s_k}) \cdot R_{t,s}\hat{\mathbf{x}}(p_{t_k}) \geq \rho\big) \wedge \big(\hat{\mathbf{z}}(p_{s_k}) \cdot R_{t,s}\hat{\mathbf{z}}(p_{t_k}) \geq \rho\big) \bigg). \tag{8}$$

vwhere $\mathrm{I}(\cdot)$ is an indicator function, $(\cdot)$ denotes the dot product between two vectors, and $\rho$ is a threshold on the angle between the corresponding axes, $0.97$ in our experiments. *Rep* measures the percentage of reference frames which are aligned, *i.e.* differ only by a small angle along all axes, between the two views. The final value of *Rep* for a given model is computed by averaging on all the pairs.

**Test-time adaptation**: Due to the self-supervised nature of Compass, it is possible to use the test set to train the network without incurring in data snooping, since there is no external ground-truth information involved. This test-time training can be carried out very quickly, right before the test, to adapt the network to unseen data and increase its performance, especially in transfer learning scenarios. This is common practice with self-supervised approaches [21].

**Datasets**: We conduct experiments on three heterogeneous publicly available datasets: 3DMatch [44], ETH [30, 11], and Stanford Views [8]. 3DMatch is the reference benchmark to assess learned local 3D descriptors performance in registration applications [44, 9, 37, 5, 11]. It is a large ensemble of existing indoor datasets. Each fragment is created fusing 50 consecutive depth frames of an RGB-D sensor. It contains 62 scenes, split into 54 for training and 8 for testing. ETH is a collection of outdoor landscapes acquired in different seasons with a laser scanner sensor. Finally, Stanford Views contains real scans of 4 objects, from the Stanford 3D Scanning Repository [8], acquired with a laser scanner.

**Experimental setup**: We train Compass on 3DMatch following the standard procedure of the benchmark, with 48 scenes for training and 6 for validation. From each point cloud, we uniformly pick a keypoint every 10 cm, the points within 30 cm are used as local surface patch and fed to the network. Once trained, the network is tested on the test split of 3DMatch. The network learned on 3DMatch is tested also on ETH and Stanford Views, using different radii to account for the different sizes of the models in these datasets: respectively 100 cm and 1.5 cm. We also apply test-time adaptation on ETH and Stanford Views: the test set is used for a quick 2-epoch training with a 20% validation split, right before being used to assess the performance of the network. We use Adam [18] as optimizer, with 0.001 as the learning rate when training on 3DMatch and for test-time adaptation on Stanford Views, and 0.0005 for adaptation on ETH. We compare our method with recent and established LRFs proposals: GFrames[24], TOLDI[42], a variant of TOLDI recently proposed in [11] that we refer to here as 3DSN, FLARE [29], and SHOT [35]. For all methods we use the publicly available implementations. However, the implementation provided for GFrames could not process the large point clouds of 3DMatch and ETH due to memory limits, and we can show results for GFrames only on Stanford Views.

**Results**: The first column of Table 1 reports *Rep* on the 3DMatch test set. Compass outperforms the most competitive baseline FLARE, with larger gains over the other baselines. Results reported in the second column for ETH and the third column for Stanford Views confirm the advantage of a data-driven model like Compass over hand-crafted proposals: while the relative rank of the baselines

changes according to which of the assumptions behind their design fits better the traits of the dataset under test, with SHOT taking the lead on ETH and the recently introduced GFrames on Stanford Views, Compass consistently outperforms them. Remarkably, this already happens when using pure transfer learning for Compass, *i.e.* the network trained on 3DMatch: in spite of the large differences in acquisition modalities and shapes of the models between training and test time, Compass has learned a robust and general notion of canonical orientation for a local patch. This is also confirmed by the slight improvement achieved with test-time augmentation, which however sets the new state of the art on these datasets. Finally, we point out that Compass extracts the canonical orientation for a patch in 17.85ms.

## 4.2 Rotation-invariant Shape Classification

**Problem formulation**: Object classification is a central task in computer vision applications, and the main nuisance that methods processing 3D point clouds have to withstand is rotation. To show the general applicability of our proposal and further assess its performance, we wrap Compass in a shape classification pipeline. Hence, in this experiment, Compass is used to orient full shapes rather than local patches. To stress the importance of correct rotation neutralization, as shape classifier we rely on a simple PointNet [31], and Compass is employed at train and test time to canonically orient shapes before sending them through the network.

**Datasets**: We test our model on the ModelNet40 [47] shape classification benchmark. This dataset has 12,311 CAD models from 40 man-made object categories, split into 9,843 for training and 2,468 for testing. In our trials, we actually use the point clouds sampled from the original CAD models provided by the authors of PointNet. We also performed a qualitative evaluation of the transfer learning performance of Compass by orienting clouds from the ShapeNet [2] dataset.

**Experimental setup**: We train Compass on ModelNet40 using 8,192 samples for training and 1,648 for validation. Once Compass is trained, we train PointNet following the settings in [31], disabling t-nets, and rotating the input point clouds to reach the canonical orientation learned by Compass. We followed the protocol described in [43] to assess rotation-invariance of the selected methods: we do not augment the dataset with rotated versions of the input cloud when training PointNet; we then test it with the original test clouds, *i.e.* in the canonical orientation provided by the dataset, and by arbitrary rotating them. We use Adam [18] as optimizer, with 0.001 as the learning rate.

Table 2: Classification accuracy on the ModelNet40 dataset when training without augmentation. NR column reports the accuracy attained when testing on the cloud in the canonical orientation provided by the dataset and AR column when testing under arbitrary rotations. Best result for each row in bold.

| | Classification Accuracy (Acc. %) | | | | | | | |
|---|---|---|---|---|---|---|---|---|
| | PointNet [31] | PointNet++ [32] | Point2Seq [20] | Spherical CNN [7] | LDGCNN [45] | SO-Net [19] | PRIN [43] | Compass + PointNet |
| NR | 88.45 | 89.82 | 92.60 | 81.73 | 92.91 | **94.44** | 80.13 | 80.51 |
| AR | 12.47 | 21.35 | 10.53 | 55.62 | 17.82 | 9.64 | 70.35 | **72.20** |

**Results**: Results are reported in Table 2. Results for all the baselines come from [43]. PointNet fails when trained without augmenting the training data with random rotations and tested with shapes under arbitrary rotations. Similarly, in these conditions most of the state-of-the-art methods cannot generalize to unseen rotations. If, however, we first neutralize the orientation by Compass and then we run PointNet, it gains almost 60 points and achieves 72.20 accuracy, outperforming the state-of-the-art on the arbitrarily rotated test set. This shows the feasibility and the effectiveness of pursuing rotation-invariant processing by canonical orientation estimation. It is also worth observing how, in the simplified scenario where the input data is always under the same orientation (NR), a plain PointNet performs better than Compass+PointNet. Indeed, as the T-Net is trained end-to-end with PointNet, it can learn that the best orientation in the simplified scenario is the identity matrix. Conversely, Compass performs am unneeded canonicalization step that may only hinder performance due to its errors.

In Figure 4, we present some models from ModelNet40, randomly rotated and then oriented by Compass. The models estimate a very consistent canonical orientation for each object class, despite the large shape variations within the classes.

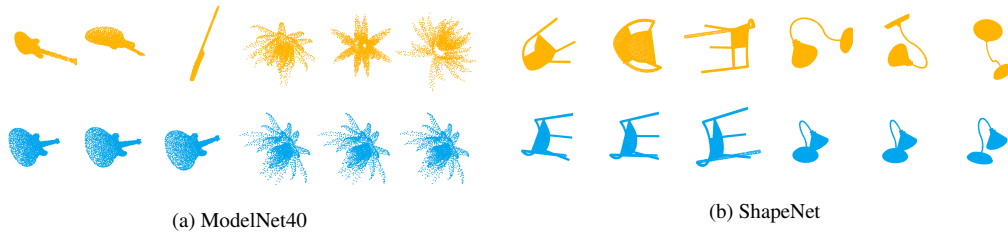

(a) ModelNet40
(b) ShapeNet

Figure 4: Qualitative results on ModelNet40 and ShapeNet in transfer learning. Top row: randomly rotated input cloud. Bottom row: cloud oriented by Compass.

Finally, to assess the generalization abilities of Compass for full shapes as well, we performed qualitative transfer learning tests on the ShapeNet dataset, reported in Figure 4. Even if there are different geometries, the model trained on ModelNet40 is able to generalize to an unseen dataset and recovers a similar canonical orientation for the same object.

## 5 Conclusions

We have presented Compass, a novel self-supervised framework to canonically orient 3D shapes that leverages the equivariance property of Spherical CNNs. Avoiding explicit supervision, we let the network learn to predict the best-suited orientation for the underlying surface geometries. Our approach robustly handles occlusions thanks to an effective data augmentation. Experimental results demonstrate the benefits of our approach for the tasks of definition of a canonical orientation for local surface patches and rotation-invariant shape classification. Compass demonstrates the effectiveness of learning a canonical orientation in order to pursue rotation-invariant shape processing, and we hope it will raise the interest and stimulate further studies about this approach.

While in this work we evaluated invariance to global rotation according to the protocol used in [43] to perform a fair comparison with the state-of-the-art method [43], it would also be interesting to investigate on the behavior of Compass and the competitors when trained on the full spectrum of SO(3) rotations as done in [10]. This is left as future work.

## 6 Broader Impact

In this work we presented a general framework to canonically orient 3D shapes based on deep-learning. The proposed methodology can be especially valuable for the broad spectrum of vision applications that entail reasoning about surfaces. We live in a three-dimensional world: cognitive understanding of 3D structures is pivotal for acting and planning.

## Acknowledgments

We would like to thank Injenia srl and UTFPR for partly supporting this research work.

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
