[Supplementary Material]

# Learning to Orient Surfaces
# by Self-supervised Spherical CNNs
# *(Supplementary Material)*

**Riccardo Spezialetti[1], Federico Stella[1], Marlon Marcon[2], Luciano Silva[3]**

**Samuele Salti [1], Luigi Di Stefano [1]**
[1] Department of Computer Science and Engineering (DISI), University of Bologna, Italy
[2] Federal University of Technology - Paraná, Dois Vizinhos, Brazil
[3] Federal University of Paraná, Curitiba, Brazil
[1]`{riccardo.spezialetti, federico.stella5}@unibo.it`
[2]`marlonmarcon@utfpr.edu.br`
[3]`luciano@inf.ufpr.br`

## 1  Learning to handle occlusions: ablation study

In this section, we study how the data augmentation carried out while training on local surface patches improves the robustness of Compass against self-occlusions and missing parts. To this end, we run an ablation experiment adopting the same training pipeline explained in the main paper at Section 3.2, without randomly removing points from the input cloud. As done in the main paper, we trained the model on 3DMatch and test it on 3DMatch, ETH, and Stanford Views. We compare Compass against its ablated version in terms of repeatability of the LRFs.

Results for 3DMatch are shown in Table 1: the performance gain achieved by Compass when deploying the proposed data augmentation validates its importance. Indeed, without the proposed augmentation FLARE performs better than Compass on this dataset. Differently, when tested in transfer learning on Stanford Views, Compass achieves state-of-the-art performance even when not using data augmentation in training, as reported in Table 2. The positive effect of augmentation is confirmed, as deploying it significantly improves the overall repeatability even on this dataset. The same observations can be made on the ETH dataset, Table 3, where, however, the gain provided by the augmentation is smaller on average. By analyzing the single scenes, we can see that augmentation is beneficial on the Gazebo fragments (both winter and summer), while detrimental on Wood scenes. We hypothesize this to be due to the different paths followed by the sensors when acquiring the two scenes, which can be seen on the dataset website[1] and are reproduced here for the reader convenience in Figure 1 : on the Gazebo scenes, the sensor undergoes several rotations and sees the same area from different viewpoints, while in the Wood scenes the path is mostly rectilinear. This means that, on the one hand, in the Gazebo scenes, occlusions are frequent due to the view point changes, and the area of overlap between two fragments often includes the border of the clouds, causing missing regions in the local patches around the keypoints: hence, data augmentation, which precisely makes the network robust to these nuisances, is effective and significantly increases repeatability. On the other hand, in the Wood scenes, the area of overlap mostly happens at the centre of the fragments and the main difference between fragments is not the acquisition angle but the acquisition distance, which does not cause occlusions or missing regions, but changes the sampling density of the cloud: therefore, the proposed data augmentation is not useful to counteract it.

`summer:home&s[]=gazebo&s[]=summer`

| (a) Gazebo Summer | (b) Gazebo Winter |
|---|---|
| (c) Wood Summer | (d) Wood Autumn |

Figure 1: Paths on ETH dataset.

Overall, these results validate our intuition that, albeit simple, the proposed data augmentation strategy effectively approximates the nuisances present in real-world dataset. Hence, it could also be successfully deployed to strengthen the robustness against occlusions and missing parts in other 3D applications, *e.g.* when learning 3D descriptors.

Table 1: LRF repeatability on the 3DMatch dataset. Best result for each row in bold.

| | | | | LRF Repeatability (*Rep* ↑) | | |
|---|---|---|---|---|---|---|
| | SHOT [5] | FLARE [4] | TOLDI [6] | 3DSN [2] | Compass (w/o aug) | Compass |
| Kitchen | 0.189 | **0.330** | 0.171 | 0.181 | 0.274 | 0.315 |
| Home 1 | 0.251 | 0.354 | 0.243 | 0.236 | 0.370 | **0.397** |
| Home 2 | 0.226 | 0.339 | 0.213 | 0.214 | 0.353 | **0.365** |
| Hotel 1 | 0.194 | **0.385** | 0.213 | 0.216 | 0.347 | 0.370 |
| Hotel 2 | 0.193 | **0.405** | 0.223 | 0.226 | 0.349 | 0.393 |
| Hotel 3 | 0.240 | 0.407 | 0.261 | 0.276 | 0.406 | **0.446** |
| Study | 0.186 | 0.351 | 0.195 | 0.192 | 0.307 | **0.356** |
| Lab | 0.220 | 0.310 | 0.198 | 0.223 | 0.360 | **0.361** |
| Mean | 0.212 | 0.360 | 0.215 | 0.220 | 0.346 | **0.375** |

Table 2: LRF repeatability on the Stanford Views dataset. Best result for each row in bold.

| | | | | LRF Repeatability (*Rep* ↑) | | | |
|---|---|---|---|---|---|---|---|
| | SHOT [5] | FLARE [4] | TOLDI [6] | 3DSN [2] | GFrames [3] | Compass (w/o aug) | Compass | Compass (adapted) |
| Armadillo | 0.127 | 0.185 | 0.156 | 0.141 | 0.168 | 0.311 | 0.340 | **0.359** |
| Buddha | 0.134 | 0.194 | 0.202 | 0.192 | 0.181 | 0.295 | 0.312 | **0.344** |
| Bunny | 0.106 | 0.379 | 0.232 | 0.172 | 0.426 | 0.358 | 0.440 | **0.463** |
| Dragon | 0.161 | 0.207 | 0.201 | 0.188 | 0.251 | 0.343 | 0.352 | **0.384** |
| Mean | 0.132 | 0.241 | 0.197 | 0.173 | 0.256 | 0.326 | 0.361 | **0.388** |

## 2 Quantitative results on 3DMatch rotated

3DMatch rotated is a synthetically rotated version of the 3DMatch dataset. This dataset has been specifically proposed to verify the invariance to rotations of the learned 3D descriptors [1], and contains only a test split. In Table 4, we show a comparison between the repeatability obtained with Compass on 3DMatch and 3DMatch rotated. Thanks to the equivariance property of the Spherical CNNs, we are able to achieve a similar performance on both datasets.

Table 3: LRF repeatability on the ETH dataset. Best result for each row in bold.

| | SHOT [5] | FLARE [4] | TOLDI [6] | 3DSN [2] | Compass (w/o aug) | Compass | Compass (adapted) |
|---|---|---|---|---|---|---|---|
| | | | LRF Repeatability (*Rep* ↑) | | | | |
| Gazebo Summer | 0.293 | **0.345** | 0.241 | 0.241 | 0.291 | 0.337 | 0.330 |
| Gazebo Winter | 0.266 | 0.268 | 0.170 | 0.196 | 0.286 | 0.292 | **0.303** |
| Wood Autumn | 0.253 | 0.210 | 0.157 | 0.174 | 0.304 | 0.288 | **0.307** |
| Wood Summer | 0.279 | 0.236 | 0.171 | 0.198 | 0.329 | 0.314 | **0.329** |
| Mean | 0.273 | 0.264 | 0.185 | 0.202 | 0.303 | 0.308 | **0.317** |

Table 4: LRF repeatability of Compass on 3DMatch and 3DMatch rotated.

| | Compass (3DMatch rotated) | Compass (3DMatch) |
|---|---|---|
| | LRF Repeatability (*Rep* ↑) | |
| Kitchen | 0.312 | 0.315 |
| Home 1 | 0.391 | 0.397 |
| Home 2 | 0.359 | 0.365 |
| Hotel 1 | 0.361 | 0.370 |
| Hotel 2 | 0.383 | 0.393 |
| Hotel 3 | 0.447 | 0.446 |
| Study | 0.348 | 0.356 |
| Lab | 0.355 | 0.361 |
| Mean | 0.369 | 0.375 |

## 3 Equivariant feature maps

In order to gain some insights into what the network learns and why it fails to correctly orient some keypoints, we studied pairs of repeatable and non-repeatable keypoints by a graphical representation of the feature maps. In particular, we studied the deepest feature map computed by Compass (before the soft argmax layer) and visualized it by plotting its top-500 bins. To represent feature map coordinates, *i.e.* rotations, in a 3D space, we used the Rodrigues vector representation because the Euclidean distance between such vectors approximates better the geodesic distance on the group manifold than the Euclidean distance between Euler angles.

In Figure 2, we consider two pairs of local surface patches and their corresponding feature maps: both patches forming a pair are extracted around the same keypoint on different fragments. The canonical pose computed for the first pair is repeatable, while the second pair represents a failure of Compass. Indeed, the feature map shown in (d), which is that computed for the patch depicted in (a), once rotated with the ground-truth transformation which aligns it with (b), is very similar to that shown in (e) and computed directly from (b). The red and green dots in (e), *i.e.* the output of Compass on (a) and (b), respectively, almost align, and result in a repeatable pair. Note how (a) and (b) are not exactly the same patch: in (a), in particular, the left-most part of the table is missing. Hence theoretical equivariance would not be guaranteed by the Spherical CNN framework alone. We ascribe the successful performance of Compass to the training loss and the augmentation we propose, which forces the network to produce similarly peaked feature maps also for slightly different input patches, like (a) and (b). The pair in (f) and (g), instead, produces feature maps that, even when rotated with the ground-truth transformation aligning the input patches, concentrate their highest values in different parts of the SO(3) space, shown by the red dot and the green dot in (j). We believe this is due to the table leg showing up only in the target cloud: legs are likely to be a robust clue across the dataset to orient keypoints sampled on tables, and its presence in the cloud down-weighs other potentially interesting features, like the mug. A more data-driven augmentation able to suppress important clues in the input cloud to make the network robust to such situations could be an interesting future work.

(a) Source cloud        (b) Target cloud

(c) Source feature map      (d) Rotated source feature map      (e) Target feature map

(f) Source cloud        (g) Target cloud

(h) Source feature map      (i) Rotated source feature map      (j) Target feature map

Figure 2: Visualization of pairs of clouds, each one extracted around the same keypoint on different fragments, and of the corresponding deepest feature maps computed by Compass for a repeatable pair (from (a) to ((e)) and a non-repeatable pair ((f) to (j)). Feature maps coordinates are represented by using the Rodrigues vector representation of rotations. Values of the feature maps are color-coded from blue to orange, the hotter the color the higher the value in a bin; the bigger red dot is the output of the network (soft-argmax). Subfigures: (a) and (f) are the source input clouds; (b) and (g) are the target input clouds; (c) and (h) are the deepest feature maps computed for the source cloud; (d) and (i) are the same feature maps, rotated by the ground-truth rotation aligning source with target; and (e) and (j) show the feature maps for the target clouds. The green dot in (e) and (j) corresponds to the red dot in (d) and (i), respectively.

# 4 Qualitative results dealing with orienting local surface patches

In this section, we provide qualitative results to show the effectiveness of Compass at computing the canonical pose for local surface patches. Given a pair of fragments, we visualize in both fragments at each point the accuracy of the estimated LRF using two different metrics. In particular, in Figure 3 we show the repeatability of the estimated LRFs (as defined in the main paper), in Figure 4 the angular distance between two rotations used as loss to train out network. In both figures, we visualize the results yielded by Compass alongside FLARE, which offers high performance across all datasets and is the second best on 3DMatch

|          (a) Compass          |          (b) FLARE          |

Figure 3: Visualization of repeatability at corresponding points of two fragments, with repeatable LRFs in green, non-repeatable ones in red and non-overlapping areas in gray. First row: a pair of fragments from Stanford Views, second row: a pair of fragments from 3DMatch. (a) and (b): results yielded by Compass and FLARE, respectively.

|          (a) Compass          |          (b) FLARE          |

Figure 4: Visualization of the angular error between the LRFs estimated at corresponding points of two fragments, with lower errors in blue, higher errors in red and non-overlapping areas in gray. First row: a pair of fragments from Stanford Views. Second row: a pair of fragments from 3DMatch. (a) and (b): results yielded by Compass and FLARE, respectively.

We can observe how Compass tends to yield larger areas in which the LRFs are accurately estimated, *i.e.* either green or blue ones, depending on the considered metric. It is worth pointing out how this is particularly evident across those challenging fragment areas affected by large missing parts in one of the two views, like, *e.g.* the left ear of the Bunny in the fragments taken from the Stanford Views dataset.

## 5    Qualitative results dealing with orienting global shapes

In this section, we provide qualitative results on the ShapeNet dataset, which complement those reported in the main paper. We stress the generalization capability of our model by adopting three different configurations to generate the training data. For this experiment, we consider only three categories: airplane, chair and lamp. The results of this study are shown in Figure 5. In the first column, (a), we present results for a *category-specific* training, *i.e.* learning to orient only one category. Thus, we train one network for each category and then we test on the test split of the same category. In (b), we present results for a category-agnostic network, *i.e.* a single model trained on samples from the three categories. Finally, in (c) we show the orientation results of the model trained according to the protocol defined in the main paper, *i.e.* transferring to ShapeNet a model trained on the ModelNet40 dataset. From these results, we observe how the canonical pose can be often correctly recovered under random rotations, and how for each triplet of rotated objects (colored in yellow) the estimated canonical pose (in blue) is consistent, even in a transfer learning strategy. Interestingly, looking at the fourth and sixth row of the (b) and (c) cases, where the model has to define a canonical orientation for more than one category at once, the canonical pose learned by the network seems to be similar across the chair and lamp categories, which have as the first principal direction the direction of gravity. This suggests that our network may generalize the concept of canonical pose across objects of different categories that share a similar geometric structure.

(a) Category-specific training       (b) Category-agnostic training       (c) Transfer learning

Figure 5: Qualitative results on ShapeNet dataset under different training strategies. Clouds in yellow represent randomly rotated input clouds and the blue ones represent those oriented by Compass. In (a), we present orientation results after training Compass with examples belonging only to a specific category from ShapeNet; in (b), the orientation results after training Compass with a training set comprising *airplanes*, *chairs* and *lamps* together; and, in (c) the orientation results from the model trained on the ModelNet40 dataset and tested on the ShapeNet dataset.

## Footnotes

[1]`https://projects.asl.ethz.ch/datasets/doku.php?id=laserregistration:gazebo_`