[Reviews · NeurIPS 2020]

Review 1

Summary and Contributions: A spherical CNN is used to predict canonical orientations for point clouds.

Strengths: Results seem better than recent state of the art methods.

Weaknesses: The task seems rather simpler than the one considered in 'Fully convolutional geometric features', ICCV 2019. The point of orienting surfaces seems to be to perform matching. But if state of the art matching can be performed without Spherical CNNs, is orienting surfaces actually a useful task? Are spherical CNNs better than non-spherical CNNs in terms of performance vs accuracy. There is no accounting for the computation cost of the method (FLOPs or run time). Table 2. I think the more interesting comparison would be the alternative methods trained with rotational data augmentation. If the only weakness with the other methods is that they need 3-degrees of freedom data augmentation, the case for Compass is a lot weaker.

Correctness: Yes

Clarity: Yes

Relation to Prior Work: Yes.

Reproducibility: Yes

Additional Feedback: L10: Ortoghonal Reply to rebuttal: There seems to be two main claims in the paper 1. That their method is useful for feature matching. The results in table 1 don't really evaluate feature matching (versus FCGF D3Feat, and related work.) They are instead optimising some intermediate objective (creating LRFs), the importance of which is not vey clear. In the rebuttal they refer to D3Feat, to explain their motivation. However, the D3Feat paper actually says '''..., as also observed in FCGF [2], we find that a fully convolutional network (e.g., KPConv [33] as used in this paper) is able to empirically achieve strong rotation invariance through low cost data augmentation, as shown in the right columns of Table 1.''' This seems to be the exact opposite to what they claim, that establishing LRFs is useful?? 2. That their method is useful for object classification. The evaluation here seems unsatisfactory as (a) they consider simple CAD models with no scanning occlusion and limited interclass variation and (b) they compare to uncompetitive baselines they trained themselves in an obviously flawed way (without appropriate data augmentation) to make an academic point about their methods rotation invariance, not to measure real world usefulness.


Review 2

Summary and Contributions: This paper presents a method to rotate 3D point sets to a canonical orientation in a self-supervised manner. The key insight is to use rotation equivariant networks, predict a canonicalizing rotation, and training them in Siamese fashion can allow canonicalization. The paper realizes this using Spherical CNNs. Experiments on two tasks show improved performance on extracting features for matching and shape classification.

Strengths: I really like the central insight of this paper. Here are the positives. - The idea of canonical rotations (called "poses" in the paper) is really neat. There is ample evidence to show that this is a useful property to have and can be used in a variety of downstream tasks. Figure 1 is great. - The paper is well motivated and the results are promising. - The idea of using spherical CNNs to achieve canonicalization is great.

Weaknesses: I would like to see this paper accepted -- but I am torn because the paper feels unfinished. It falls short in theoretical/experimental rigor, and writing quality. I list my questions, comments, and suggestions below which I hope can help address the the shortcomings. - First, the paper refers to canonical orientation as "pose" which is misleading. The term "pose" usually refers to position and orientation, i.e. SE(3) transformations whereas what the paper achieves is equivariance to SO(3). This should be clarified. - The description of spherical CNNs mostly made sense as they follow [6]. But I found it a bit hard to follow section 3.2 without reading it multiple times. Part of the reason is because of some abuse of notation in equation (4). g^{-1}(.) in reality is an SO(3) rotation but the way it is defined makes it look like it is a point cloud (since g(.) maps form P-->SO(3), g^{-1} should map from SO(3) --> P). It may make sense to change this notation. Moreover, the text explanations in 135 are not very clear and succinct. - Section 3.2 does not make it clear that equivariance enables canonicalization only because of the property of spherical CNNs that they output features in SO(3). Any neural network without this output domain will not be canonicalizing (unless they explicitly predict rotation). To me, this is the *main insight* of the paper and is not highlighted anywhere. - For the loss function in (6), is it applied once for each layer or once for the whole network? Why? - How does the method handle permutation equivariance for the points? Is a standard lexicography adopted for the sphere cells? What about the permutation equivariant components of PointNet in the shape classification experiment? - In the local surface patches experiment, how does the overlap between patches help? Does more overlap improve results? - I would have loved to see visual and quantitavative results of the canonical patches discovered by the network.

Correctness: Mostly yes. Some notation abuse in equation (4) onwards for g^{-1}(.)

Clarity: The quality of writing in the paper can be significantly improved. For instance, the Standard Flow in line 126 does not make sense because the conversion of point clouds to spherical signals is described later in line 161 onwards. Section 3 is hard to read and can benefit from re-writing.

Relation to Prior Work: Not enough. The paper is missing discussion of the following papers on canonicalization. - C3DPO: Canonical 3D Pose Networks for Non-Rigid Structure From Motion, Novotny et al. 2019 - Normalized Object Coordinate Space for Category-Level 6D Object Pose and Size Estimation, Wang et al. 2019 - Multiview Aggregation for Learning Category-Specific Shape Reconstruction, Sridhar et al. 2019

Reproducibility: Yes

Additional Feedback: Thanks for the rebuttal. The writing can be improved -- overall I am leaning slightly on the positive side.


Review 3

Summary and Contributions: The paper presents an approach for self-supervised learning of a NN that maps identical (up to noise and occlusion) 3D point clouds into identical canonical orientations. Having such a NN is demonstrated to enable two tasks: learning rotation invariant 3D descriptors and rotation-robust point cloud classification. The basic idea is to learn filters of a NN $g$ defined on directional signals. First, the point cloud is converted into a direction-distance representation. The first two dimensions are angles, the third one a distance. Now training takes two copies of those patches, the original one $V$ and a rotated one, $T$, transformed by a known rotation $R$. Now $g$ is learned to output an activation map that has a max at one bin $i,j,k$. That bin maps to a certain rotation (orientation and rotation around it). The loss now asks, that the transformation between those two maximal activations on the two point cloud copies is exactly R. Applying this transformation, the two point clouds can be aligned, not maybe semantical (as in a cup being upright), but consistently. This step then is a preprocess (or learned jointly with) another task, such as classifying the point cloud or to compute descriptors for matching. Results on classic ShapeNet etc benchmark show a marked increase in performance, in particular for randomly rotated objects.

Strengths: -- Strong technical idea -- Good exposition of a difficult (I thought) topic -- Good results for a focused analysis, executed fairly -- I have not seen soft argmax (which is easily confused with softmax in the start) before and find it a useful tool

Weaknesses: -- Not easy to understand -- I could not understand the relation of SO(3) and this orientation/distance binning. I would not see what it really i --it appears this is just a spherical grid-- and what the relation of i,j,k, is to a specific rotation. That seems to follow from the spherical CNN's properties, but it did not get across for me -- Also in the analysis, some simpler competitors like PCA could be compared to. PointNet already has a means to output a rotation to work in a consistent (local) frame. Why does it fail? Probably, cause it does not use spherical CNNs? The analysis could even think about a downgraded version of PointNet that also does not use this transformer. -- The protocol for data augmentation in respect to occlusions was surprising: I would have expected some simulated occlusions, as in Blendsor, but instead some concentric shells that drop some points are used. This seems not to be very representative of true occlusions. But it could also be seen as a strength that even with such a primitive occlusion model, results are good. So with a better model, that might be easy to include, results would become even better. -- I was unsure why this need spherical CNNs. What is wrong with a pipeline that would just voxelize the point cloud, and encode it to a rotation matrix or (unit) quaternion? Or more similar: voxelize the point cloud and produce a SO(3) activation map, where each bin is one rotation. Which property of spherical CNNs is crucial where? Right now the choice of Spherical CNN appears principled as using a point net, voxels or a graph CNNs.

Correctness: I did not spot any mistakes. Some smaller typos. L28 it says PointNet achieves rotation invariance by random rotations. That is correct, but not complete. There is also a step in the pipeline to rotate -- very much like what is done here-- into a canonical frame. It just does not seem to work as well as what is suggested here.

Clarity: Yes. I had difficulties following, but maybe not the papers fault. -- Some typeset matrices as \mathsf -- I am always confused by functions like g returning a matrix in Eq. 4. When it then says ^-1, is it the inverse of the matrix or the inverse of the mapping g? Here it is the inverse of the matrix, but maybe some brackets or re-ordering could help? Wikipedia agrees, how function inversion and multiplicative inversion ARE confusing notations. Fig. 2 could be linked better to Eq. 4 and 5. I see T and S, but not R, R-star, g etc. In fact, all the funny blocks of the architecture do not matter much to me, if only i would see all relevant operations applied to all relevant point clouds. In particular, what is compare to what in the end for computing a loss?

Relation to Prior Work: All I am aware of.

Reproducibility: Yes

Additional Feedback:


Review 4

Summary and Contributions: This paper presents a methodolody, named compass, that learns canonical orientation of 3D objects and surfaces by self-supervised spherical CNN. The basic idea is to learn a canonocial SO(3) matrix by Spherical CNN on two sets V,T, supervised by a known rotation matrix R. In terms of contribution, it provides a effective and robust orientation learning method for various 3D vision tasks.

Strengths: 1. The paper is written clearly and well motivated. The theoretical claims, mainly grounded on Spherical CNN, is sound and clear. 2. Extensive experiments are conducted to demonstrate the effectiveness of the proposed method, and they are explained in detail. And judging by the qualitative results, especially the transfer learning result, the robustness of Compass is promising.

Weaknesses: 1. The proposed idea is mainly grounded on Sperical CNN (2018). It seems like that the current Compass is a simple combination of Spherical CNN and a self-consistency angular loss, plus some learning techniques like deleting points to handle occlusion. From this perspective, the theoretical novelty of this paper is unclear. I would like authors to clarify more on the originality w.r.t. the previous works. 2. This one is more like a confusion. In your 2nd experiment, where you replace the T-Net with Compass followed by PointNet, I would assume that Compass can provide a good canonical transformation far better than a simple T-Net. But in Table 2, the Compass + PointNet cannot beat simple PointNet in NR scenario, I would like the author to give some insight about it.

Correctness: I do not see obvious error in the methodology.

Clarity: The paper is well written and easy to follow.

Relation to Prior Work: Like I suggest in weakness part, I would like the author to clarify more on the theoretical originality w.r.t. the previous works.

Reproducibility: Yes

Additional Feedback: Post-Rebuttal: I've read the rebuttal. I think the central idea is interesting, but the concern about empirical evaluation is only partially addressed in the rebuttal. So I decide to keep my initial score for the submission.

[Author Response · NeurIPS 2020]

**R1**: We agree that defining a canonical orientation for local patches is mainly aimed at descriptor matching. However, we propose a more general framework that can also be adopted to orient whole objects and perform rotation-invariant shape classification. Moreover, as recently shown in Bai *et al.* in "D3Feat: Joint Learning of Dense Detection and Description of 3D Local Features." (CVPR 2020), while FCGF performs very well when trained and tested on the 3DMatch dataset, it suffers from a large drop in performance in transfer learning towards the ETH dataset. This weakness in transfer learning is particularly critical for supervised methods, such as FCGF, because it limits applicability only to datasets for which the ground truth is available. One reason for this performance drop is likely to be the handling of rotation invariance by data augmentation, which may hardly generalize to unseen datasets. Indeed, 3DSN [10], which achieves rotation invariance by a LRF, outperforms the competitors by a considerable margin in transfer learning on ETH. Based on these considerations, we believe that it is not yet been established that rotation-invariant descriptor matching can be solved without orienting surfaces. Thus, as vouched by its large performance gain with respect to existing LRFs across different datasets, we believe that Compass is a principled and useful contribution that can improve the performance of learned feature descriptors relying on LRFs.

Compass extracts the canonical orientation for a patch in 17.85ms. We will add this information to the revised version.

In the evaluation in Table 2, we follow the standard protocol used in [39] to perform a fair comparison. We believe that this kind of evaluation provides important insights as it highlights whether learned methods can generalize or not to unseen rotations. We agree with the reviewer that it would also be interesting to investigate on the behavior of the competitors when trained on the full spectrum of SO(3) rotations, but we could not run such experiment in the limited time available to complete the rebuttal, as it requires re-training all competitors listed in Table 2. We will mention it as future work and highlight the importance of this complementary assessment in the conclusions.

**R2**: We used the more general term "pose" to refer to canonical orientation as achieving translation invariance is usually trivial, but we agree that this use may be misleading. As suggested, we will use only the term *orientation*.

(Pointed out also by **R3**) We agree that the notation in (4) should be changed from $g^{-1}(\mathcal{V})$ to $g(\mathcal{V})^{-1}$, since inversion is applied to the output of function $g$, *i.e.* the learned rotation matrix. We will modify it in the final version of the paper.

We agree that the domain of Spherical CNNs feature maps is key and we will better highlight it in the final version.

Since we seek for one rotation, the loss function in (6) is applied once, and only to the last layer of the network.

The input of our network is a spherical signal that is invariant to permutations of the input data. More details about it can be found in [39]. As for PointNet, we use the original implementation provided by the authors.

We experimentally verified that a larger overlap between fragments improves the repeatability of Compass on local patches. We will add this insight to the discussion of the experimental results.

**R3**: The output of a spherical correlation is a signal living in SO(3). In particular, each feature map is a cube where each cell, indexed by $i, j, k$, represents an element of SO(3), *i.e.* a rotation.

As suggested, we trained a version of PointNet without the T-Nets. The performance in test is: NR: 88.49; AR: 8.35.

The use of Blendsor would have required generation of a cumbersome off-line training dataset. We instead used an on-the fly data augmentation where occlusions can be randomly generated across epochs: since it proved effective, as demonstrated by the ablation study, we consider its simplicity a positive aspect, as suggested also by the reviewer.

We rely on Spherical-CNNs because equivariance to rotations is crucial to satisfy (5) and they are equivariant to SO(3) by construction. Conventional neural networks do not possess this property.

We will update Figure 2 including all the symbols adopted in the definition of the methodology. The loss compares the max entries of the final features maps, which corresponds to rotations, as explained above.

**R4**: We present the first machine learning approach to orient point clouds. Differently from previous handcrafted solutions, no geometrical cues are adopted to design a repeatable canonical orientation, while we leverage the equivariance of Spherical CNNs and show that a fully data-driven approach is feasible and, indeed, more effective than SOTA solutions. It is worth pointing out that this problem can not be tackled by supervised learning as there is no unique manner to define the ground-truth (i.e. a repeatable canonical orientation). Thus, we propose a self-supervised formulation where the network is able to discover the best suited canonical orientation based on training data.

Unlike T-Net, Compass is not trained end-to-end with PointNet. In the simplified scenario where the input data is always under the same pose (NR), a canonicalization step is useless and can only lower performance by injecting noise due to its errors. T-Nets trained jointly with PointNet can instead learn that the best orientation in this scenario is the identity matrix. Yet, Compass could be easily modified to be trained end-to-end with the down stream task: this is an interesting future work we would like to explore.

[Meta-Review · NeurIPS 2020]

This paper received overall positive ratings from three reviewers. However, the submission also received slightly conflicting comments in almost all aspects : novelty, writing and experiments. Some valid negative points concerning the empirical evaluation were raised by R1, and those were weighted amongst all reviewers during the discussion period. Taking all this into consideration, we decided on an acceptance as poster. I strongly encourage the authors to take the reviewer's feedback into account when preparing the camera-ready version, notably those concerning clarity and writing.